# Influence of Alkali Treatment on the Mechanical, Thermal, Water Absorption, and Biodegradation Properties of *Cymbopogan citratus* Fiber-Reinforced, Thermoplastic Cassava Starch–Palm Wax Composites

**DOI:** 10.3390/polym14142769

**Published:** 2022-07-06

**Authors:** Zatil Hafila Kamaruddin, Ridhwan Jumaidin, Rushdan Ahmad Ilyas, Mohd Zulkefli Selamat, Roziela Hanim Alamjuri, Fahmi Asyadi Md Yusof

**Affiliations:** 1Fakulti Kejuruteraan Mekanikal, Universiti Teknikal Malaysia Melaka, Hang Tuah Jaya, Durian Tunggal 76100, Malaysia; zatilhafila@gmail.com (Z.H.K.); zulkeflis@utem.edu.my (M.Z.S.); 2German-Malaysian Institute, Jalan Ilmiah Taman Universiti, Kajang 43000, Malaysia; 3Fakulti Teknologi Kejuruteraan Mekanikal dan Pembuatan, Universiti Teknikal Malaysia Melaka, Hang Tuah Jaya, Durian Tunggal 76100, Malaysia; 4School of Chemical and Energy Engineering, Faculty of Engineering, Universiti Teknologi Malaysia, Johor Bahru 81310, Malaysia; ahmadilyas@utm.my; 5Centre for Advanced Composite Materials (CACM), Universiti Teknologi Malaysia, Johor Bahru 81310, Malaysia; 6Faculty of Tropical Forestry, Universiti Malaysia Sabah, Jalan UMS, Kota Kinabalu 88400, Malaysia; 7Malaysian Institute of Chemical and Bioengineering Technology, Universiti Kuala Lumpur, Alor Gajah 78000, Malaysia; fahmiasyadi@unikl.edu.my

**Keywords:** *Cymbopogan citratus* fiber, palm wax, starch, biodegradation, alkali treatment

## Abstract

In this study, thermoplastic cassava starch–palm wax blends, reinforced with the treated *Cymbopogan citratus* fiber (TPCS/ PW/ CCF) were successfully developed. The TPCS were priorly modified with palm wax to enhance the properties of the matrix. The aim of this study was to investigate the influence of alkali treatments on the TPCS/PW/CCF biocomposite. The fiber was treated with different sodium hydroxide (NaOH) concentrations (3%, 6%, and 9%) prior to the composite preparation via hot pressing. The obtained results revealed improved mechanical characteristics in the treated composites. The composites that underwent consecutive alkali treatments at 6% NaOH prior to the composite preparation had higher mechanical strengths, compared to the untreated fibers. A differential scanning calorimetry (DSC) and a thermogravimetric analysis (TGA) indicated that adding treated fibers into the TPCS matrix improved the thermal stability of the samples. The scanning electron microscopy (SEM) demonstrated an improved fiber–matrix adhesion due to the surface modification. An increment in the glass transition temperature (T_g_) of the composites after undergoing NaOH treatment denoted an improved interfacial interaction in the treated samples. The Fourier transform infrared spectroscopy (FTIR) showed the elimination of hemicellulose at wavelength 1717 cm^−1^, for the composites treated with 6% NaOH. The water absorption, solubility, and thickness swelling revealed a higher water resistance of the composites following the alkali treatment of the fiber. These findings validated that the alkaline treatment of CCF is able to improve the functionality of the *Cymbopogan citratus* fiber-reinforced composites.

## 1. Introduction

Plastics that originate from petroleum-based chemicals are the driving force behind industrialization, and their strength and versatility have helped to raise our standard of living through their use in a variety of applications. However, the accumulation of plastic waste has caused severe environmental problems, as well as the devastation of ecosystems [1]. Therefore, managing plastic waste continues to be a major environmental challenge, and transitioning to biodegradable plastics could help to alleviate some of these problems. Many researchers have been exploring new biodegradable plastics in order to develop fully biodegradable plastics with acceptable qualities [2,3].

Starch is often regarded as the most promising feedstock for biodegradable plastics owing to its benefits, which are as follows: its abundant sources, low cost, biodegradability, renewability, and processability [1,4,5,6,7]. Starch, obtained from a wide range of botanical sources (cereals, legumes, rhizomes, and tubers), is a readily available raw material that can be used in bioplastic applications [8]. On the other hand, starch has also been found to have the ability to create rigid materials, notably thermoplastic starch, which was also discovered in prior research [6,9]. In general, thermoplastic starch exhibits similar physical properties as synthetic polymers (e.g., being tasteless, translucent, and odorless), [1]. However, besides its great advantages, pure thermoplastic starch also has several limitations, including low mechanical strength, long-term stability, and water resistance, limiting its potential applications [5,7]. Thus, it is frequently required to modify thermoplastic starch to make this material practical in real applications. The combination of thermoplastic starch with other natural polymers appears to be a potential technique for improving the drawbacks of this biopolymer, while still preserving the biodegradability of the material [10]. In our study, the mechanical, thermal, and physical characteristics of modified thermoplastic starch, which was derived from cassava starch and palm wax, were reported [4]. The findings demonstrated substantial improvements in the mechanical properties of the thermoplastic cassava starch–palm wax blend, after the palm wax was incorporated [4].

It is noteworthy that reinforcing thermoplastic starch with a natural fiber is an intriguing approach to addressing the drawbacks of thermoplastic starch. Using these fibers as fillers or as reinforcements in blends has received more attention, especially for applications that are price-driven and require large volumes. Some studies have been performed on the addition of natural fibers, e.g., kenaf [11], cogon grass [12], sugar palm [7], kapok [5], and jute [5], into the thermoplastic starch matrix. The majority of the studies essentially emphasized increasing the mechanical and thermal characteristics of thermoplastic starch. In addition, the incorporation of natural fibers into thermoplastic starch material yielded improved water resistance [5,13].

*Cymbopogan citratus* is one of the aromatic members of the Poaceae family that has recently been drawing attention in the field of natural fiber composite manufacturing, due to its potential as a polymer reinforcement [14,15]. The attractive characteristics of natural fibers are their abundance in nature, renewability, low cost, and their high specific modulus, in addition to the fact that they are lightweight compared to the synthetic fibers [12,16]. Despite its many advantageous properties, the use of this fiber in composite applications is limited by its poor adhesion to the polymeric matrix, as well as its poor dimensional stability, which is due to the fact that fibers are generally hydrophilic, whereas matrices are hydrophobic [17]. On the other hand, like all other natural fibers, *Cymbopogan citratus* fibers also have poor interfacial adhesion between the polymer matrix and the raw natural fibers, and lower thermal resistance is often not suitable for the intended applications. In general, the celluloses contained in the *Cymbopogan citratus* fiber are the main structural fiber components, whereas lignin and hemicelluloses are crucial in determining the fiber’s characteristics. The approximate composition of the *Cymbopogan citratus* fiber is as follows: cellulose (37.56%), hemicelluloses (29.29%), lignin (11.14%), and ash substances (4.28%) [18]. The hydrophilic hydroxyl (–OH) groups are present in cellulose, hemicellulose, and lignin as part of their structure. The presence of easily accessible hydroxyl groups in amorphous lignin and hemicellulose allows water molecules to permeate the surface of the fiber [19]. Consequently, the hydroxyl groups in cellulose (in the amorphous area) interact with water molecules and remain in the fiber’s structure, resulting in high moisture absorption and in preventing close intermolecular contact with hydrophobic polymer matrices [20]. As a result, the fiber and matrix have poor interfacial bonding. Hence, a preliminary chemical modification of the fiber surface is frequently required for interfacial bond strength and compatibility improvements. 

Alkalization, bleaching, and acetylation are chemical treatments that are commonly used to enhance the matrix–fiber adhesion [21]. These treatments reduce the hydroxyl group number in the amorphous region; eliminate a specific amount of lignin, hemicellulose, and other non-cellulosic components from the fiber’s surface; and increase the reaction of cellulose with the binder materials by exposing the cellulose structure, resulting in better-purified cellulose [17]. It has been proven in numerous research studies that using chemical treatments on natural fibers enhances the mechanical and thermal characteristics of composites. Fiora et al. [17] reported that the mechanical strength of kenaf fiber increased when it was treated with 6% of NaOH. Meanwhile, Yousif et al. [20] noted a 36% improvement in the flexural strength of kenaf–epoxy composites by using 6% of NaOH, which was a result of the hydrophilic components’ removal from the fiber, allowing for improved compatibility with the matrix. Furthermore, Aydin et al. [22] performed an experimental investigation that utilized higher NaOH concentrations (10%, 20%, and 30%) on short flax fiber–poly-lactic acid composites and further revealed a gradual decrement of the tensile modulus with the rising NaOH concentration. The decline in the tensile modulus was associated with the extensive fiber damage from the increased alkali concentrations. In terms of thermal properties, Liu et al. reported that increasing the alkali concentration resulted in an improved thermal stability of Indian grass fibers by increasing the temperature at the maximum rate of decomposition following the alkali treatment. This was associated with the elimination of lignin and hemicellulose following the application of NaOH [23].

Despite the various efforts made to discover the alkali treatment’s effects on the properties of some natural fiber composites, none were found to use the *Cymbopogan citratus* fiber to reinforce modified thermoplastic cassava starch. In addition, very limited information about the thermo-mechanical properties of such thermoplastic biocomposites have been obtained. In this work, the influence of the fiber surface treatment on the properties of the ensuing composites was determined by experimental investigations into the composites’ morphological characteristics as well as into their thermal, physical, mechanical, and biodegradation properties. 

## 2. Materials and Methodology

### 2.1. Materials

In this work, the *Cymbopogan citratus* plant leaves were obtained from a farm in Beranang, Selangor, Malaysia. The cassava starch (food-grade) was supplied by Antik Sempurna Sdn. Bhd (Selangor, Malaysia). Evergreen Engineering Sdn. Bhd. (Selangor, Malaysia) provided the required chemicals, including sodium hydroxide (NaOH) pellets of 98% purity, acetic acid, and analytical grade glycerol (99.5% purity). The refined palm wax (analytical grade) was purchased from Green & Natural Industries Sdn. Bhd., (Selangor, Malaysia).

### 2.2. Preparation of Cymbopogan Citratus Fiber

The fiber from *Cymbopogan citratus* leaves used in this work was extracted using the water-retting method, by soaking the leaves in a container of water for 4 weeks. The individual fiber bundles were isolated from the retted leaves and then cleansed under running water. Next, the fibers were dried at 100 °C for 5 h in an oven to eliminate the excess moisture. The dried fibers were then cut into 1 cm lengths and stored in a zip-locked plastic bag.

### 2.3. Alkaline Treatment

In this study, the dried raw *Cymbopogan citratus* fiber treatment was accomplished with three different compositions of aqueous NaOH (3, 6, and 9 wt.%), each for a period of 60 min and left under agitation at room temperature (28 °C). Generally, this alkali treatment facilitated the elimination of great amounts of hemicellulose and surface impurity components from the fibers’ structure. Next, the treated fibers were cleaned under running water, then pH neutralization was performed using acetic acid to eliminate any excess alkali deposits formed on the fibers’ surface. Lastly, the treated fibers underwent drying at 25 °C for 48 h and further oven drying at 60 °C for 24 h to remove any remaining moisture traces from the fibers.

### 2.4. Sample Preparation

The thermoplastic cassava starch fabrication was conducted in accordance with our previous work [24]. The ratio of starch: glycerol: palm wax was fixed at 65:30:5 (wt.%). Following this step, the mixture was then blended using a Dry Mixer, Panasonic (Shah Alam, Selangor, Malaysia) at 1200 rpm at room temperature for 5 min. The resultant mixture then underwent thermo-pressing for 30 min at a temperature of 150 °C, using a 40HC-B Technopress, Plastic Hydraulic Moulding Press (Selangor, Malaysia) with a 10-tonne load, the yielding plates having 3 mm thickness. Similar processes were used for the modification of TPCS/PW with treated CCF. The matrix’s property alterations were carried out by incorporating treated CCF, where 50 wt.% of matrix was used. Before conditioning, the prepared samples were immediately placed in a silica gel-filled desiccator.

### 2.5. FT-IR Analysis 

The FT-IR analysis of the treated and untreated TPCS/PW/CCF composites was conducted using FTIR machine JASCO FTIR-6100 Spectrometer (Tokyo, Japan), equipped with an ATR platinum diamond crystal and the manufacturer’s OMNIC software, to examine the functional groups’ changes in their fiber surfaces. All spectra within 4000–400 cm^−1^ range were recorded with a resolution of 4 cm^−1^, for a total of 32 scans per measurement. The ATR crystal was cleaned with ethanol prior to each measurement, and the background spectra were collected and removed automatically from the recorded spectrum. An anvil was used to press the sample down into the ATR diamond crystal before the test was performed in order to create close proximity between the ATR diamond crystal and the sample. The anvil pressure setting of the ATR-FTIR was subject to variation and was not reproducible throughout all samples due to the manual adjustment. OMNIC software was used to control and correct the baseline of measurements after the correct band of spectrum was selected.

### 2.6. Scanning Electron Microscopy (SEM)

The morphology of the fractured tensile samples’ untreated and treated TPCS/PW/CCF were examined using a scanning electron microscope (SEM), model JEOL JSM-6010 Plus (Tokyo, Japan), at 10 kV acceleration voltage. All samples were covered with gold prior to being subjected to SEM observation. 

### 2.7. Tensile Testing

The tensile characteristics of the specimens were investigated following the ASTM D-638 standard [25]. An INSTRON 5969 model Universal Testing Machine, manufactured by INSTRON (Noorwood, MA, USA), with 50 kg load cell. The crosshead speed of the machine was kept constant at 5 mm/min, this was used to measure tensile modulus and strength, as well as elongation. The tensile properties were estimated as the average of the obtained values after five replications of the measurements.

### 2.8. Flexural Testing

The flexural test of untreated and treated TPCS/PW/CCF composite was carried out using an INSTRON 5969 Universal Testing Machine, manufactured by INSTRON (Noorwood, MA, USA), in accordance with ASTM D-790 [26], with 50% relative humidity. The samples, with dimensions of 130 mm (L) × 13 mm (W) × 3 mm (T), were prepared, and the analysis was performed using five (5) replicates, which were evaluated at 23 ± 1 °C temperature.

### 2.9. Impact Testing

Izod impact tests of untreated and treated TPCS/PW/CCF composite samples were conducted in accordance with the ASTM D256 standard [27]. The Izod impact test was conducted at 23 ± 1 °C and RH of 50 ± 5%. The samples with the following dimensions: 60 mm (L) × 13 mm (W) × 3 mm (T), were prepared, and five replicates of each sample were tested in a Ray-ran impact tester, manufactured by Nuneaton, United Kingdom. The cross-sectional area of the specimens, as well as the impact energy, were considered while determining the impact strength Equation (1), as follows: Impact strength = Impact energy (J)/area (mm^2^)(1)

### 2.10. Thermogravimetric Analysis (TGA)

The thermal properties for untreated and treated TPCS/PW/CCF composites were analyzed using a thermogravimetric analyzer, Mettler Toledo AG, Analytical (Schwerzenbach, Switzerland). The samples were analyzed under a dynamic nitrogen atmosphere surrounding, at 50 mL/min flow rate, with temperatures ranging from 25 to 600 °C, at a constant heating rate (10 °C/min^−1^). A sample pan containing 5–15 mg of a composite was heated. TGA graphs of weight loss percentage versus temperature were used to identify the weight loss.

### 2.11. Differential Scanning Calorimeter (DSC)

The thermal behavior of the treated TPCS/PW/CCF composite was evaluated by differential scanning calorimetry (DSC) equipment (Universal V3.9A TA, Instruments New Castle, PA, USA). Each composite sample was weighed to a 5 mg precision and placed in an aluminum sample pan with an air-tight lid during the procedure. The samples were heated to temperatures ranging from 35 to 250 °C, at a continuous scanning rate of 10 °C/min. The thermogram measurements for the DSC cell were generated using transfer temperatures, which were flushed with nitrogen gas to sustain an inert atmosphere.

### 2.12. X-ray Diffraction (XRD)

The XRD patterns of the untreated and treated TPCS/PW/CCF composite samples were investigated using X-ray diffractometer (Rigaku, Tokyo, Japan), employing CuKα radiation (λ = 1.5406 nm) generated at 40 mA and below 40 kV, respectively. Crystallinity index CI (%) of each sample was determined according to Equation (2), where I_002_ and I_am_ are the peak intensities of crystalline and amorphous materials, respectively. Equation (2) is as follows:(2)$$\mathrm{CI}=\frac{I_{002}-I_{\mathrm{am}}}{I_{002}}\times 100\%$$

### 2.13. Density

The density of the treated TPCS/PW/CCF composite was carried out according to the ASTM D1895. A total of five samples were prepared (10 mm × 10 mm × 3 mm) and dried for 24 h in the oven at 105 °C. Afterwards, the samples were placed in a desiccator filled with granulated silica gel and a weighing balance and electronic densimeter were used to weigh the samples and determine their volume. The density value was determined according to Equation (3), as follows:(3)$$\mathrm{Density}\;(g/{\mathrm{cm}}^{3})=\frac{\mathrm{Mass}\;\left(g\right)}{\mathrm{Volume}\;\left({\mathrm{cm}}^{3}\right)}$$

### 2.14. Moisture Content

Five samples (10 mm × 10 mm × 3 mm) were prepared for moisture content analysis. All of the samples were placed for moisture removal in an oven at 105 °C for 24 h. The weights of the samples taken from the oven were measured before (M_i_) and after (M_f_) the heating process in order to compute the moisture content. The moisture content was computed using Equation (4), as follows:(4)$$\mathrm{Moisture}\;\mathrm{content}\;(\%)=\frac{M_{i}-M_{f}}{M_{i}}\times 100$$

### 2.15. Water Absorption

Five samples (10 mm × 10 mm × 3 mm) were placed to dry for 24 h in an air-circulating oven at 105 °C ± 2 temperatures to eliminate any remaining moisture from the samples. The samples being tested were completely immersed in water at room temperature (23 ± 1 °C) for 2 h. For water absorption calculation, the mass of the samples before (W_i_) and after immersion (W_f_), and the water absorption of the samples was computed according to Equation (5), as follows:(5)$$\mathrm{Water}\;\mathrm{Absorption}\;(\%)=\frac{W_{i}-W_{f}}{W_{i}}\times 100$$

### 2.16. Thickness Swelling

Five test specimens were used for thickness swelling test using similar parameters for the testing as described in Section 2.13. The thickness of each sample was recorded before (T_i_) and after (T_f_) immersion of the samples in water for 2 h, using a Mitutoyo digital vernier with 0.01 cm precision. The thickness swelling percentage values were evaluated using Equation (6), as follows: (6)$$\mathrm{Thickness}\;\mathrm{swelling}\;(\%)=\frac{T_{i}-T_{f}}{T_{i}}\times 100$$

### 2.17. Moisture Absorption

Moisture absorption study for treated TPCS/PW/CCF composites was performed in a closed humidity chamber at room temperature of 25 ± 2 °C and relative humidity of 75 ± 2% to analyze the samples’ moisture absorption behavior. Prior to conducting the test, five (5) samples with dimensions of 10 mm × 10 mm × 3 mm were prepared and oven-dried for 24 h at 105 °C ± 2. The samples were weighed before (Wi) and after absorption (W_f_) Equation (7) for a specified duration until a stable weight was attained. The moisture absorption of samples was determined using Equation (7), as follows:(7)$$\mathrm{Moisture}\;\mathrm{Absorption}\;(\%)=\frac{W_{f}-W_{i}}{W_{i}}\times 100$$

### 2.18. Water Solubility

The water solubility of the samples was studied following the method reported by Chaireh et al. [28]. Prior to the water solubility investigation, five samples measuring 10 × 10 × 3 mm were cut and dried in an oven at 105 °C ± 2 for 24 h. The initial dry matter of each piece was recorded as W_i_. Next, each sample was immersed in 30 mL distilled water while constantly agitated. After 24 h of immersion, the remaining section of the sample was taken out of the beaker and wiped using a filter paper to remove any remaining water on the surface. Finally, all samples were dried again at 105 °C ± 2 temperature for 24 h to obtain the final weight of the sample, denoted as W_f_. Equation (8) was used to calculate the water solubility of the samples, as follows:(8)$$\mathrm{Water}\;\mathrm{Solubility}\;(\%)=\frac{W_{i}-W_{f}}{W_{i}}\times 100$$

### 2.19. Soil Burial

A soil burial test was conducted in compliance with the procedure described in Jumaidin et al. [12]. The five samples (10 mm × 10 mm × 3 mm) were prepared and buried at 10 cm depth in soil and regularly moistened with distilled water. The ambient temperature and relative humidity (RH) for the analysis were 26 ± 4 °C and 76 ± 4%, respectively, and the pH of the soil was 6.5. The samples were wrapped in iron mesh before being buried in the soil, allowing the degraded materials to be eliminated, while still permitting microorganisms and moisture access. Before testing, all samples were dried at 105 °C for a continuous 24 h, and the weights were recorded in order to compute the initial weight, which was denoted by M_i_. All samples were buried in the soil for predetermined periods ranging between 2 and 4 weeks. Following that, the samples were cautiously removed from the soil at a certain period, and the impurities were softly washed with distilled water to remove any remaining contaminants. Afterward, the degraded samples were dried for 24 h in an oven at 105 °C before being reweighed to obtain the final weight, M_f_. The biodegradability of the samples was determined via a comparison of the weight loss before and after burial, which was calculated using Equation (9), as follows:(9)$$\mathrm{Weight}\;\mathrm{loss}\;(\%)=\frac{M_{i}-M_{f}}{M_{i}}\times 100$$

### 2.20. Statistical Analyses

The statistical analyses of the experimental results were conducted using the analysis of variance (ANOVA) function in SPSS software. The Duncan test was also used to compare the mean values between the two groups at a significance level of 0.05 (*p* ≤ 0.05). Results were summarized using the average and standard deviation for each sample (SD). 

## 3. Results

### 3.1. FT-IR Analysis

Figure 1 displays the spectrum obtained from the FT-IR analysis of the untreated and treated TPCS/CCF composites, showing that virtually identical patterns were obtained, with minor differences in the case of the treated sample. The presence of cellulose, hemicellulose, and lignin components in the fiber structure was determined by the peak response found at specific wavelengths, between 400 and 4000 cm^−1^. A wide peak observed at approximately 3282 cm^−1^ was identified by the characteristic band for the –OH stretching of untreated fibers. Additionally, the alkali treatment shifted the band from 3282 to 3296 cm^−1^, respectively, and it was more intensified compared with the untreated composite. This shift might be associated with the hydrogen bonding reduction in the cellulosic hydroxyl groups, which led to an increased –OH concentration. This phenomenon indicated a decrease in the hydrophilic nature of the fiber and more reactive –OH groups were exposed to react with the matrix [29]. A prominent absorption band (approximately between 2800 and 2900 cm^−1^) for the treated and untreated TPCS/CCF, respectively, might be linked with the alkyl C–H stretching vibration (both symmetrical and asymmetrical) in the cellulose and hemicellulose components in natural fibers [30]. This adsorption band decreased slightly following the alkali treatments, which is attributable to the hydrogen bonding breakdown between the –OH groups of the cellulose and hemicellulose molecules [31]. A minor noticeable peak, at approximately 2349 cm^−1^, on the TPCS/CCF composite indicated the presence of wax compounds on the fiber surface [32]. 

Meanwhile, the absorption band at 1717 cm^−1^ was visible and found in the untreated TPCS/PW/CCF composite and was treated by NaOH with a concentration of 3 wt.%. The vibration band was related to the carbonyl (C=O) stretching vibrations of the carboxyl and acetyl groups in hemicelluloses [33]. However, it was discovered that the band was absent in the fibers treated with NaOH concentrations of 6 wt.% and 9 wt.%, indicating the dissolution of lignin and hemicellulose during alkali treatment [34]. This finding is consistent with a prior study, which found that alkali treatment can eliminate hemicelluloses at certain NaOH concentrations [35]. In this study, it seemed the NaOH concentration at 6% was sufficient to remove the hemicellulose present in the treated fibers.

The absorption bands at 1643 and 1242 cm^−1^ indicated the C=C and C–O stretching of acetyl groups in the lignin components, respectively [21]. However, the intensity of the bands was no longer visible for the treated TPCS/PW/CCF composite samples, thus, indicating partial lignin elimination from the fibers following the alkali treatment [36]. Cellulose demonstrated an obvious peak at 1028 cm^−1^ due to the polysaccharide C–O stretching vibration, whereas β-glycosidic links between the monosaccharides and C–OH bending were represented by smaller and broader intense peaks, at 881 and 605 cm^−1^, respectively [37]. Overall, it can be summarized that the alkali treatments possessed a degree of reactivity in eliminating the majority of the lignin and hemicellulose components from the fibers, changing the hydrophilic *Cymbopogan citratus* fibers to hydrophobic, thus, improving the compatibility of both the fibers and the matrix.

### 3.2. Scanning Electron Microscope (SEM)

Scanning electron microscopy (SEM) is an effective method of examining the longitudinal surface of the fiber and the fractured morphology of the TPCS/PW/CCF composites. Figure 2 presents the corresponding SEM micrographs of both the untreated and treated samples at various NaOH concentrations. From observation, it was found that after NaOH treatment, significant differences in the fiber morphologies in the treated samples were observed, compared to the untreated samples. Figure 2a presents the SEM micrograph of the longitudinal untreated fiber surface. The wax, lignin, and other considerable impurities were present on the untreated fiber surface, as can clearly be observed. Furthermore, hemicelluloses and waxes led to the appearance of white patches on the untreated fiber, significantly decreasing the interfacial bonding properties of the fiber throughout the composite development [38,39]. Meanwhile, Figure 2b–d shows the longitudinal surface micrographs of fibers treated with 3, 6, and 9 wt.% of NaOH, respectively. A significant alteration in the morphological fiber surface structure was observed after the NaOH treatment. It was noticed that the varied percentages of NaOH caused the white patches and other impurities to be eliminated after treatment, as presented in Figure 2b–d. The surface roughness of the fiber surface was improved after the lignin, hemicelluloses, and wax removal from the surface of the fiber by alkalization [32]. According to the previous studies, rough surface morphologies are typical for treated fibers because hemicellulose and other surface impurities have been removed [40]. Thus, alkali treatment can cause significant modifications in the fiber surface morphology, which can enhance the wettability of the fiber. Furthermore, the removal of surface contaminants on fibers increased both the mechanical interlocking and the bonding reaction, which is advantageous for fiber–matrix adhesion [41]. However, after being treated with a high concentration of NaOH (9 wt.%), a cleaner surface was noticed and the pores on the fiber surface were more visible. This might be associated with the removal of fatty components from the fiber surface [40]. Nevertheless, if there are more pores present than the specified level, the mechanical characteristics of individual fibers may be compromised [42].

Meanwhile, Figure 2e demonstrates the tensile fractured surface morphology after the tensile test for the untreated TPCS/PW/CCF composites. Figure 2g,h shows the micrograph for the treated TPCS/PW/CCF biocomposites, which were treated with NaOH 3, 6, and 9 wt.%, respectively. In general, the CCF is dispersed in both transverse and longitudinal directions in the TPCS matrix, as shown in Figure 2e–h. From the observation of the fracture surface of the NaOH-treated composites, a rougher surface than in the untreated TPCS/PW/CCF composites was found. This was probably associated with the improvement in the bonding strength between the matrix and the fiber [17]. As seen in Figure 2g, it was demonstrated that good adhesion between the matrix and fiber was achieved through the alkali treatment, which effectively assisted the fibers’ dispersion and the stress transfer throughout the entire composite structure, improving the final mechanical characteristics of the materials. In addition, the NaOH treatment extracted a specific amount of wax, lignin, hemicellulose, and artificial contaminants from the surface of the fibers, while simultaneously increasing the overall roughness of the fiber surface, producing an improved mechanical interlocking between the matrix and the fibers [42]. In addition, a recent study found that fibers’ surface modification improved the fiber–matrix adhesion, which, in return, improved the composites’ mechanical performance [17]. 

However, when the samples were treated with a 9 wt.% NaOH solution (Figure 2h), it was revealed that the fiber surface was damaged by the high alkali concentration, leading to surface topographical deterioration. Nevertheless, the SEM image appeared to show more holes and grooves, indicating that the alkaline treatment was capable of removing very large amounts of soluble substances from the sample [40]. The reduction in the fiber–matrix interfacial bonding might be associated with the high concentration of NaOH, which induced fiber damage and led to decreased strength in the TPCS/PW/CCF composites [29].

### 3.3. Tensile Testing

A uniaxial tensile test was used to evaluate the mechanical properties of the sample. Figure 3 demonstrates the effects of the NaOH treatment with various chemical concentrations on the tensile strength and modulus, as well as the percent elongation at break of the TPCS/PW/CCF biocomposites. From the results, Young’s modulus and the tensile strength of the treated TPCS/PW/CCF biocomposites showed significant improvement (*p* < 0.05), to the maximum values of 6574.88 MPa and 19.89 MPa, respectively, at 6 wt.% NaOH concentration. This result demonstrated that the treated fibers experienced slight increments in their tensile strength and modulus, unlike the untreated TPCS/PW/CCF biocomposites. 

In general, several factors could be responsible for the increased tensile modulus and strength of the treated TPCS/PW/CCF composites. Firstly, this enhancement can be ascribed to the extraction of the major portions of lignin, hemicelluloses, and other unwanted components from the fibers, facilitating the fibrils rearrangement along the tensile deformation direction, yielding an increased tensile strength [43]. From the results, the optimum tensile strength was reported at the 6 wt.% NaOH treatment, reaching the value of 19.89 MPa with an increment of 21.65%, compared to the untreated TPCS/PW/CCF biocomposites. These findings were in agreement with our previous study, which showed that an alkali treatment with 6 wt.% NaOH solution resulted in an enhanced tensile strength of the fiber-reinforced composite [29]. Second, a good interface bonding and compatibility between the matrix and fiber after the treatment resulted in the development of a rough surface on the fibers, which, in turn, led to improved adhesion between the matrix and the fiber [44]. In addition, the fiber surface roughness increased, which produced a better mechanical interlocking and led to less fiber–matrix debonding. The existence of fiber breakage in the SEM image of the tensile fracture surface verified this, implying that there was better adhesion at the interphase. 

However, when the NaOH concentration was increased by 9 wt.%, the tensile strength and the modulus demonstrated a decreasing trend to 13.24 MPa and 2558.81 MPa, respectively, and this was the lowest tensile strength and modulus produced out of all the NaOH concentrations used. This might be related to the excess alkali treatment of the TPCS/PW/CCF composites dissolving the hydrogen bonds in the cellulose and lignin structure, resulting in the softening of the fibers [29]. This removal process might have been a contributing factor to the modulus and crystallinity reductions in the fiber, which might have affected the mechanical properties of the samples [37]. Despite the positive results on the tensile modulus and strength shown by the NaOH concentration of 6 wt.%, when they were treated with a greater alkali solution, the Young’s modulus decreased and the strain at break increased (9 wt.% NaOH). This could be associated with the alkali treatment causing hydrogen bonds to be eliminated in the cross-linked networks of the lignin and cellulose structures, which may have yielded the reduction in Young’s modulus and extended elongation [36]. This finding is consistent with a prior study that examined the effects of alkali treatment on the tensile characteristics of the abaca fiber [36].

### 3.4. Flexural Testing

Figure 4 presents the effect of the NaOH treatment on the flexural strength and modulus of the treated and untreated TPCS/PW/CCF biocomposites. The flexural strength, also known as the modulus of rupture, is a useful parameter for estimating the efficiency of composites when they are subjected to structural loads. In general, the effect of the NaOH treatments on the flexural properties of the treated TPCS/PW/CCF composites exhibited a similar pattern to the effect of the NaOH treatments on the tensile properties of the treated TPCS/PW/CCF composites. From the results, the TPCS/PW/CCF biocomposites with 6 wt.% NaOH treatment demonstrated the maximum flexural strength (30.06 MPa) and modulus (7526.33 MPa). The treated TPCS/PW/CCF composites that were subjected to the 6 wt.% NaOH treatment revealed improvements (7.5% in flexural strength and 21.1% in flexural modulus), as compared to the untreated TPCS/PW/CCF composite samples. The role of NaOH in terms of the fiber characteristics’ modifications and the increased interfacial bonding between the fiber and the matrix could explain these improvements in the flexural strength and modulus [42]. However, the results also demonstrated a reduction in the improvement trend of the samples’ flexural performance when the NaOH treatment concentration was increased to 9 wt.%. The flexural strength and modulus of the TPCS/PW/CCF composites were decreased by 136.94% and 266.97%, respectively, unlike the untreated TPCS/PW/CCF composites in the presence of a high concentration of alkali for the 9 wt.% NaOH treatment. These findings were consistent with those of Vilay et al. [45] for incorporating treated bagasse fiber-reinforced unsaturated polyester composites.

### 3.5. Impact Testing

The Izod impact tests can be used to determine the strength of materials. The amount of energy absorbed by a material during a fracture can be determined through this testing. The Izod impact strength of the untreated and treated *Cymbopogan citratus* fiber-reinforced thermoplastic cassava starch–palm wax composite samples are shown in Figure 5. From the impact testing, we saw that increasing the alkaline concentration (0 to 6 wt.%) demonstrated an improvement in the impact strength by 230.37% (*p* < 0.05). This result denoted a significant improvement in the instance of impact testing after alkali-treating the composite. Moreover, the alkali treatment improved the fiber–matrix bonding, producing a composite with improved bending and crack propagation resistance. However, the higher alkali concentration of the composite led to a decline in the impact strength. This phenomenon denoted the decline in the impact strength with a higher percentage of alkali solution concentrations (6 to 9 wt.%), which could be ascribed to the higher rigidity of the material resulting in more brittle properties, thus, reducing its ability to absorb impact energy [46]. As a result, it was demonstrated that the percentage of the alkali solution concentration employed for the treated samples had a significant impact on the impact strength behavior. Table 1 summarizes the analysis of variance (ANOVA) findings of the tensile, flexural, and impact properties, which can be used for statistical interpretation. According to the results, the mean values of the mechanical strength and modulus between the composite levels differed from one another. These differences were statistically significant (*p*-value < 0.05, respectively).

### 3.6. Thermogravimetric Analysis (TGA)

TGA is used to evaluate mass loss, thermal decomposition, and the thermal stability of materials in the temperature range where they can be utilized, until they degrade noticeably. Figure 6a displays the TGA and DTG of the untreated and treated TPCS/PW/CCF composites. The thermal stability of the samples was studied in a temperature range of 30 to 600 °C. The thermal degradation curves of the untreated and treated TPCS/PW/CCF composites were noticeably different. Besides, the TG curves of all the treated TPCS/PW/CCF composites exhibited the same pattern, despite the fact that the samples underwent treatments with different NaOH concentrations. The elimination of loosely-bound water and low-molecular-weight compounds attributed to the initial stage of degradation, which occurred below 100 °C and in which it lost approximately 3.87–5.35% of its weight [32]. The degradation phases that took place between 100 to 200 °C were attributed to the water evaporation and the glycerol decomposition, respectively [5]. Meanwhile, the maximum degradation took place at approximately 300 °C and was assigned to the depolymerization and devolatilization of the starch carbon chains [5]. The degradation of the chemical components in the TPCS/PW/CCF biocomposites, namely the hemicellulose, took place at a temperature of approximately 190 °C, while the degradation of the cellulose occurred between 290 and 360 °C, respectively [47]. In this case, the treated TPCS/CCF composite exhibited less weight loss compared to the untreated composite, in up to 100 °C temperature. This might be ascribed to the higher moisture content in the untreated *Cymbopogan citratus* fiber [48]. On the other hand, the maximum weight losses of approximately 60.03 and 54.49%, were revealed for the untreated and treated CCF-reinforced TPCS/PW composites. Meanwhile, the lignin degradation and the final decomposition of the cellulose presented a broad peak throughout the range 280–500 °C [49]. 

In addition, the treatments significantly impacted the thermal degradation behavior of the TPCS/PW/CCF composites, causing a decrease in the temperatures at which the thermal degradation occurred, from 369–336 °C. This could be associated with the fact that lignin, hemicellulose, and volatiles were eliminated in the alkali treatment and these peaks correlated with the α-cellulose decomposition [50]. It was noticeable that the untreated TPCS/PW/CCF composites had the lowest final residue percentage, followed by the treated TPCS/PW/CCF composites, which were treated with 3, 6, and 9 wt.% NaOH, respectively, with the values of 18.49, 20.89, and 21.92%, respectively. This alkaline treatment positively affected the thermal degradation behavior of the treated TPCS/PW/CCF composites, as was evidenced by the increased final residue percentage, which could be interpreted as good thermal stability [16]. This, according to several studies, is related to the extraction of lignin from the natural fiber after treatment [51]. Nevertheless, Ndazi et al. reported that this final residue was attributed to the condensed polycyclic aromatic structure formation of the cellulose as the solid residue underwent a chemical rearrangement and transformation [50].

The DTG analysis of the untreated and treated CCF-reinforced TPCS/PW composites are presented in Figure 6b. The degradation temperature of each component of the TPCS/PW/CCF composites was responsible for the DTG curve peaks. A difference between the untreated and treated TPCS/PW/CCF composites was observed, with the untreated TPCS/PW/CCF composites exhibiting three peaks, whereas all of the treated samples only revealed two peaks. Due to the existence of water molecules in the hemicelluloses of the untreated fiber, the first peak was higher than in the other treated TPCS/PW/CCF composites. The first degradation took place in the range of 25–100 °C for both the untreated and treated TPCS/PW/CCF composites. The organic materials began to degrade at this temperature. This was attributed to the presence of moisture in the fiber, which evaporated at this temperature. The second degradation peak of the untreated TPCS/PW/CCF composites occurred between 250–350°C, whereas for the treated TPCS/PW/CCF composites it occurred between 250–300 °C. The second stage of decomposition is represented by the hemicellulose decomposition that occurred at higher temperatures [52]. The degradation of cellulosic components was observed in the third degradation step, which occurred only for the untreated TPCS/PW/CCF composites, at 320–385 °C. However, the third degradation step did not take place in the other types of treated TPCS/PW/CCF composites as their cellulosic component percentage was reduced due to the efficient alkali treatments, which were used as reported by Zhang et al. [53]. Overall, the lower degradation peaks for the treated TPCS/PW/CCF biocomposites indicated that some components, e.g., lignin and hemicellulose, were removed from the fiber during the alkali treatment process.

### 3.7. Differential Scanning Calorimeter (DSC)

The untreated and treated TPCS/PW/CCF composites were analyzed using a differential scanning calorimetric study. The glass transition temperature (T_g_), was an important parameter in this research as it helped us to understand the structure and the interactions between the TPCS matrix and the fibers. The T_g_ denotes the transformation of polymeric materials from a glassy to a rubber state [54]. Table 2 shows the corresponding value of the T_g_ with the treated and untreated TPCS/PW/CCF composites, where the T_g_ was determined from the heating scan. The treated composites exhibited an increase in the T_g_, compared with the untreated composites, which yielded an increment in the T_g_ from 123.9 to 128.5 °C. This could be related to the decreased mobility of the matrix chains, which would result in a higher T_g_ value. This behavior was justified by the improved interfacial adhesion achieved by the alkali treatment [55]. According to the findings, the maximum T_g_ value was recorded for the 6 wt.% NaOH-treated TPCS/PW/CCF composites at 128.5 °C (the highest T_g_ value), indicating that this is the optimal alkali concentration for improved fiber–matrix interface adhesion. The results obtained were in agreement with the results from the previous study on the effect of fiber surface treatment on the characteristics of poly (lactic acid)–ramie composites [56]. Nevertheless, comparable observations were previously noted on the treated ensete stem fiber-reinforced polyester that showed an increasing trend of the T_g_ value at 5.0% NaOH concentrations, due to the stronger fiber–matrix bonding. However, a decreased T_g_ value was obtained for the composites with a higher alkali concentration (7.5% NaOH) due to the hemicellulose and lignin eliminations at the interfibrillar region, making the fiber less dense and brittle [57].

### 3.8. X-ray Diffraction (XRD)

X-ray diffraction is a practice that provides valuable information in the analysis of sample crystallinity [58]. The X-ray diffractograms of the untreated and treated TPCS/PW/CCF composites (for various alkali concentrations) is shown in Figure 7. As observed in the diffractograms, all the composites showed two primary reflections, equivalent to 2θ values of approximately 17 and 22°, respectively. The low angle reflection (17°) was a broad-spectrum intensity, whereas the high angle reflection (22°) was sharp and intense. These reflections were ascribed to the amorphous fraction (I_am_) and the crystalline fraction (I_002_), which are parts of the composite, respectively. The alkali treatment of the samples resulted in a considerable rise in the intensities of the (I_am_) and (I_002_) crystallographic planes, as seen in Figure 7. The crystallinity index of the TPCS/PW/CCF composites was computed using Equation (2) and presented in Table 3.

The crystallinity index of the untreated composites was determined to be 37.7%, whereas the 3 wt.% NaOH-treated composite had a crystallinity index of 40.3%. This improvement was due to the incomplete eliminations of amorphous hemicellulose, wax, and non-crystalline components from the sample [59]. Meanwhile, the crystallinity index of the 6 wt.% NaOH-treated composite was 47.9%, due to a decrease in the amorphous content and the rearrangement of the crystalline region [60]. Nevertheless, the crystallinity index of the composite treated with 9 wt.% NaOH, which was 45.2%, did not show any further improvement. The observed pattern demonstrated the increment in the fraction of crystalline cellulose in the fiber that underwent an alkali treatment with up to 6 wt.% of NaOH, and which subsequently declined because of the effect of the high concentration of NaOH in the alkaline treatment on the cellulose structure [37]. This result was also verified by the FT-IR analysis, as presented in Figure 1. A similar improvement in the crystallinity index after the alkali treatment was found [59,60,61] and explained by eliminating the non-cellulosic components, which allowed for the better packing of the cellulose chain.

### 3.9. Density

Figure 8 shows the density of the untreated and treated TPCS/PW/CCF composites. In general, it was found that increasing the NaOH solution’s concentration significantly (*p* < 0.05) reduced the density value of the TPCS/PW/CCF composites. The untreated TPCS/PW/CCF composites demonstrated a density of approximately 1.25 g/cm^3^, meanwhile the treated TPCS/PW/CCF composites demonstrated a slight decrease in density values, ranging from 1.24 to 1.20 g/cm^3^, respectively. This finding might be associated with the eradication of the wax, hemicellulose, lignin, oil, and surface impurities from the fiber surface by the alkali treatment [41]. Additionally, the primary cell wall of the fiber was damaged as a result of the decreasing total weight of the fiber core and, thus, the lowering of its density value [62]. Moreover, a reduction in the density of the treated TPCS/PW/CCF composites following the increased NaOH concentrations could also be due to an increment in the porosity of the fiber after the treatment, compared with the untreated TPCS/PW/CCF composites [40]. Wong et al. [62] studied the effect of alkali treatment on bamboo fibers, where increasing the concentration of the NaOH aqueous solution was revealed to decrease the density, indicating a higher alkalization effect.

### 3.10. Moisture Content

The moisture contents of the untreated and treated TPCS/PW/CCF composites with various alkali concentrations are shown in Figure 9. In general, slight decrements in the moisture contents were evidenced by the treated composites of 6.56%, 5.40%, 4.76%, and 4.43%, respectively. Nevertheless, the moisture content of the untreated composite that showed the highest value might be associated with the hydrophilicity of untreated fibers, which contain hydroxyl groups in their structure. Moreover, this significant moisture content reduction in the TPCS/PW/CCF composites was found when the alkali concentration was increased from 3 to 9%. This phenomenon might be attributed to the strong interfacial bonding between the matrix and the fibers that halted the water movement to the interface, thus, they were free to bind to moisture. This finding corresponded with the discoveries of past studies on the moisture content behavior of treated sugar palm-reinforced sugar palm–starch matrix composites [7]. 

### 3.11. Water Absorption

The water absorption of the untreated and treated TPCS/PW/CCF composites were measured, and the variations are shown in Figure 10. From Figure 10, obviously, the treatment of the TPCS/PW/CCF composites with the alkali solution reduced the water absorption of the samples from 12.57 to 5.27%, respectively. The untreated TPCS/PW/CCF composites exhibited the highest water absorption, of 12.57%, when immersed in distilled water for 2 h. The higher water uptake for the untreated TPCS/PW/CCF composite might be ascribed to the high hydrophilic nature of the untreated *Cymbopogan citratus* fiber to TPCS matrix. On the other hand, the presence of the hygroscopic characteristics of the hemicelluloses in untreated fibers increased the fibers’ water absorption tendency, causing the fiber to expand when submerged in water [30]. However, the hydrophilic nature of the fiber was reduced with alkali treatment, as proven by the lower percentage of water absorption of the treated TPCS/PW/CCF composites, when compared to the untreated TPCS/PW/CCF composites. The presence of hemicellulose and lignin on the surface of the *Cymbopogan citratus* fibers can be clarified by the alkali treatment that removed a certain number of extractives, such as pectin, chemical components, and contaminants [16]. In fact, alkali treatments could eradicate the hydroxyl group on the surface of natural fibers, reducing the water absorption of TPCS/PW/CCF composites [63]. Consequently, this response might enhance the adhesion between the fiber and the TPCS matrix. Therefore, water absorption can be reduced due to the upgraded interfacial holding quality. The results were in accordance with the published research of Demir et al. [64] on the effect of surface treatment on luffa fiber-reinforced polypropylene. Atiqah et al. [16] also reported similar results in treated sugar palm–glass fiber-reinforced thermoplastic polyurethane, which effectively lessened the water uptake of the composite.

### 3.12. Thickness Swelling

The swelling ratio of the untreated and treated TPCS/PW/CCF composites was investigated to determine the ability of the samples to preserve their structural integrity in an aqueous environment. Figure 11 shows the rate of the thickness swelling of the samples, which continued to decrease significantly (*p* < 0.05) following further increases in the NaOH concentrations from 3 to 9 wt.%, respectively. The untreated TPCS/PW/CCF composite demonstrated the highest percentage of thickness swelling (4.32%), while the treated TPCS/PW/CCF - 9 wt.% showed the least percentage of thickness swelling (2.95%). This effect might be attributed to the untreated sample that did not bond well with the TPCS matrix. Thus, the large discrepancy between the fiber and the TPCS matrix allowed a rapid passage of water molecules into the composite. The composite swelled fast when water was absorbed by the fibers, resulting in a maximum water absorption thickness swelling rate, after 2 h [65]. Apparently, when the untreated TPCS/PW/CCF composite was immersed in the NaOH solution, some of the pectin, sugars, lignin, and other organic materials were dissolved [66], and the quantity of elements dissolved out increased as the concentration of sodium hydroxide (NaOH) in the solution increased. Thus, the samples became more voided, and the thickness swelling rate was presumably reduced [67].

Moreover, this decreasing swelling ratio value could be associated with the treatment with NaOH that induced the shrinking of the micropores and collapsed the fibers’ capillaries [68]. As a result, the fibers will not retain water, and the amount of absorbed water will be reduced. The thickness swelling will be reduced to a certain extent and will essentially remain constant in this manner. The result corresponded to a study by Hafidz et al. [68], who also observed a decrease in the swelling value of treated palm oil fiber-reinforced unsaturated polyester composites. 

### 3.13. Moisture Absorption

Figure 12 displays the moisture absorption curve of the untreated and treated TPCS/PW/CCF composites during 11 days of storage at 25 ± 2 °C, with 75 ± 2% relative humidity. Overall, all the TPCS/PW/CCF composites exhibited a similar pattern for moisture absorption, with a rising storage duration. The curve shows that moisture absorption was faster for a period of time and then slowed as the storage time increased. After nine days of storage, the saturation rate was achieved without any further increment in the moisture absorption being observed, indicating that the composites had reached an equilibrium moisture content with their environments (reached a plateau). 

As depicted in Figure 12, it is shown that the alkali treatment reduced the moisture absorption of the composite materials. The composites’ moisture absorption decreased considerably with the rising alkali concentration. The untreated TPCS/PW/CCF composites showed the highest moisture absorption, compared to the treated composites. The existence of suspended hydroxyl and polar groups in various parts of the untreated fibers have caused moisture absorption to be considerably higher and have caused the interface bonds between the natural fibers (hydrophilic) and the matrix (hydrophobic) to become lower, compared to the treated composites. Thus, the reduction in moisture absorption and the hydrophilic characteristic of the composite could be accomplished through an alkali treatment of the TPCS/PW/CCF composite, which resulted in the increased hydrophobicity of the samples [29,69]. After 11 days, a gradual moisture uptake reduction, with a rising alkaline concentration was noted for the TPCS/PW/CCF composites - 3, 6, and 9 wt.%, respectively. This result was consistent with the moisture absorption behavior of the rice and einkorn wheat husks-reinforced poly(lactic acid) biocomposite [70].

### 3.14. Water Solubility

The water solubility of the untreated and treated TPCS/PW/CCF composites is displayed in Figure 13, indicating the materials’ water resistance when immersed and continuously stirred in water. The untreated composites’ water solubility was higher than the treated ones. The water solubility of the samples was significantly (*p* < 0.05) improved as the water resistance and solubility values declined from 20.80 to 9.93%, respectively. The untreated TPCS/PW/CCF composites demonstrated the highest percentage of water solubility (20.80%). For untreated fibers, the higher solubility could be associated with the presence of hydroxyl groups in the cellulose, accessible by water [71]. Moreover, the existence of a high hydrophilic hemicelluloses content in the untreated composites raised their water solubility tendency, causing the composites to swell when immersed in water [29]. 

The hydrophilicity of the composite was lowered through the NaOH treatment, which led to the prohibiting of water from the substrate of the composite, as revealed by the solubility percentage of the NaOH-treated composites, which were less in comparison with the untreated composites. In contrast, the solubility percentage of a composite might be ascribed to several aspects, e.g., a high void content in the composite, interfacial bonding between the matrix and fibers, and the amount of fiber loading, could be responsible for this behavior. A similar finding was observed in the investigation of flax fiber-based composites, in which treated flax fibers reduced the solubility of the sample [72].

### 3.15. Soil Burial

The biodegradability level of the composite was assessed by determining the weight loss (%) of the samples after two and four weeks of soil burial. Figure 14 illustrates the percentage weight loss for untreated and alkali-treated TPCS/PW/CCF composites. It is clear that a longer burial time resulted in more weight loss in all composites, indicating a greater number of microbes in the materials. Furthermore, this might be associated with the physical characteristics of the untreated fibers, which absorbed more water than the treated fibers [73]. This left the composite more exposed to microbial organisms’ attacks, which targeted the untreated samples in the presence of water as the medium.

Overall, the alkali treatment of the composites slowed their degradation rates when in soil. Among the composites, the highest weight loss was 26.53 and 32.01% for the untreated composite, after two and four weeks of burial, respectively. This phenomenon occurred due to the properties of the untreated fibers, which were more hygroscopic than the treated ones, making them more visible, thus, promoting microbial activities. According to Siakeng et al. [74], the higher weight loss of the untreated composites than that of the alkali-treated composites could be ascribed to the poor fiber–matrix adhesion, which quickened the degradation rate of the composites. Therefore, the lower weight loss of the treated composites might be related to the alkali treatment that led to the reduced hydrophilicity nature of the fibers, meaning that there was less moisture absorption from the soil. This, consequently, led to the slower degradation rate of the alkali-treated composites [75]. This finding was in agreement with the modified jute fiber-reinforced hybrid biocomposite developed in a previous study, which incorporated polylactide–polycaprolactone blends [76].

## 4. Conclusions

In this study, the mechanical and thermal properties and the biodegradability of TPCS–palm wax biocomposites, reinforced by untreated and treated *Cymbopogan citratus* fiber have been successfully evaluated. The mechanical test of an alkali-treated *Cymbopogan citratus* fiber, pre-soaked with 6 wt.% NaOH solution, demonstrated a higher mechanical strength, at 19.9 Mpa, 30.0 Mpa, and 13.3 Mpa for tensile, flexural, and impact strength, respectively. This suggests there was a good fiber–matrix adhesion and an effective stress transfer from the matrix to the fiber. In view of the morphological structure, the longitudinal surface study of the treated fiber revealed that the surface became rough after being exposed to high alkali concentrations. An increasing trend in the crystallinity index values were demonstrated for the treated TPCS/PW/CCF biocomposites, which is in agreement with their relatively good mechanical properties. The FT-IR analysis of the composites demonstrated the removal of hemicelluloses, lignin, and other surface contaminants from the fiber surface. The treated composites showed significant improvements in their water resistance in all the tests and the soil burial tests demonstrated good biodegradability in the untreated composites, at the highest weight loss. Overall, TPCS-reinforced treated *Cymbopogan citratus* fiber, at 6 wt.% NaOH, improved the mechanical, thermal, and physical properties of the composites. This finding can expand their prospective applications as promising biodegradable composites.

## Figures and Tables

**Figure 1 polymers-14-02769-f001:**
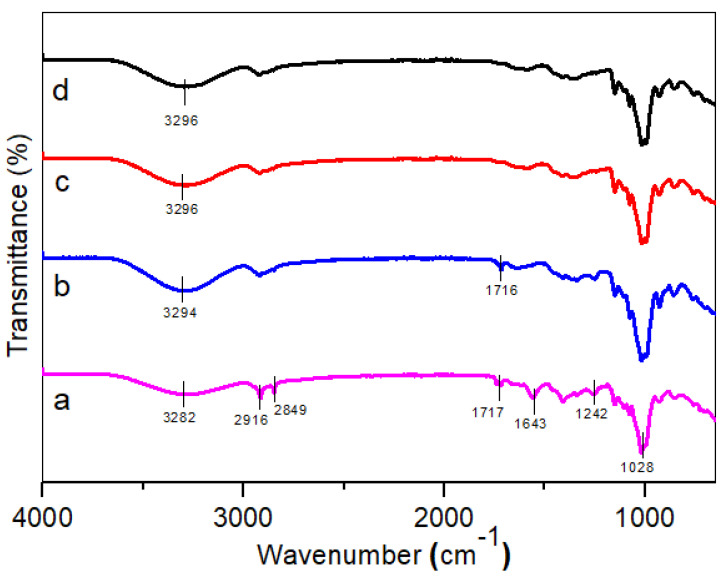
FT-IR spectra of (a) untreated, (b) 3 wt.%, (c) 6% wt.%, and (d) 9 wt.% of TPCS/PW/CCF composites.

**Figure 2 polymers-14-02769-f002:**
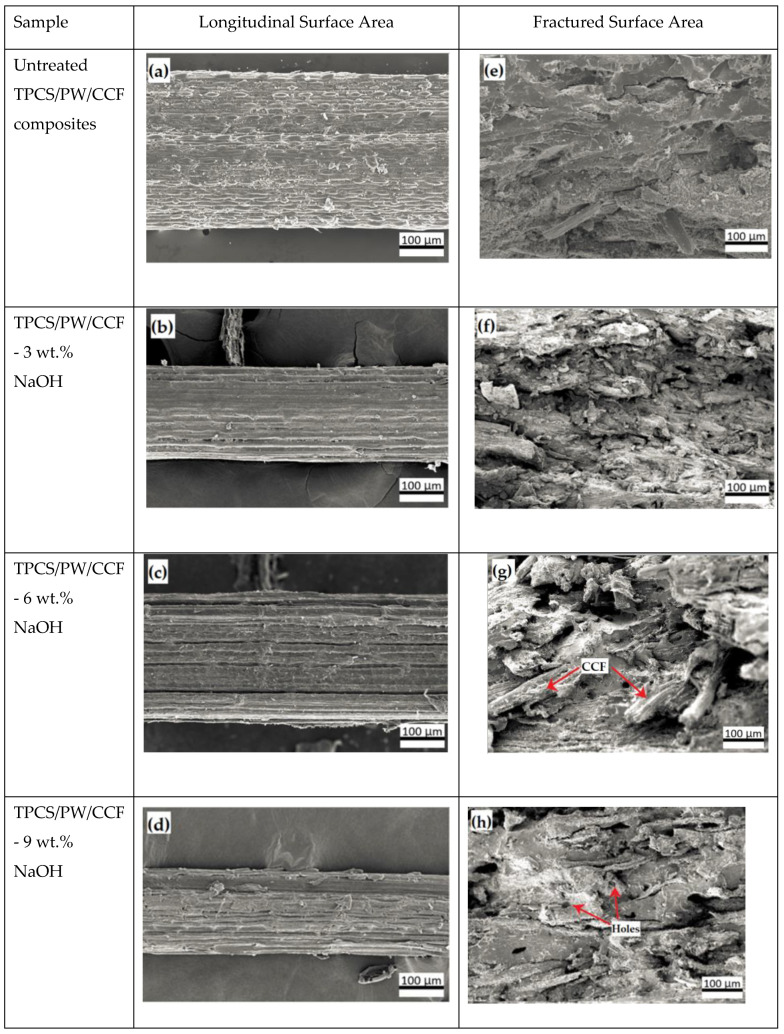
SEM images of untreated and treated TPCS/PW/CCF biocomposites from longitudinal surface and fractured surface views.

**Figure 3 polymers-14-02769-f003:**
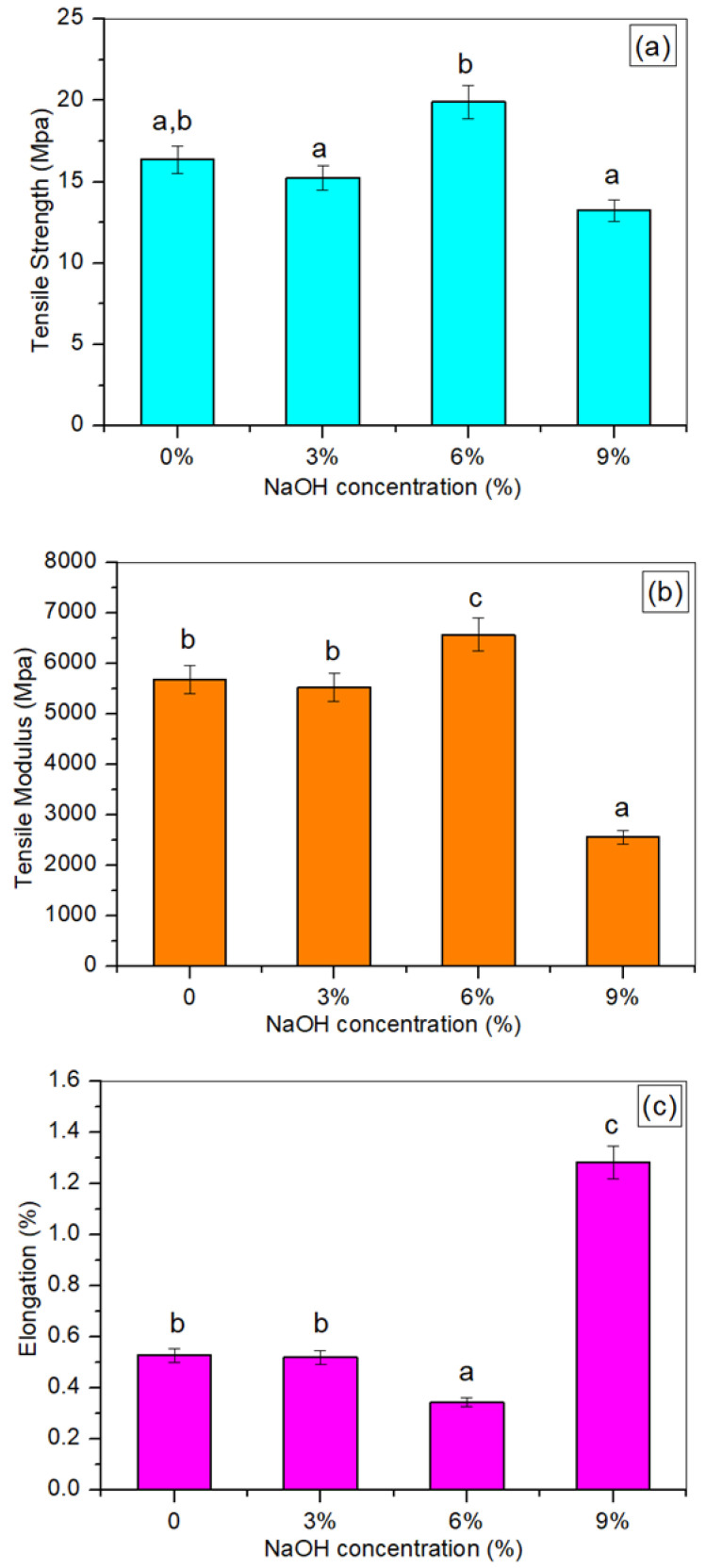
The (**a**) tensile strength, (**b**) tensile modulus, and (**c**) elongation at break of untreated and treated TPCS/PW/CCF composites at different NaOH concentrations. Letters a, b, and c is indicates the group of data for statistical analysis.

**Figure 4 polymers-14-02769-f004:**
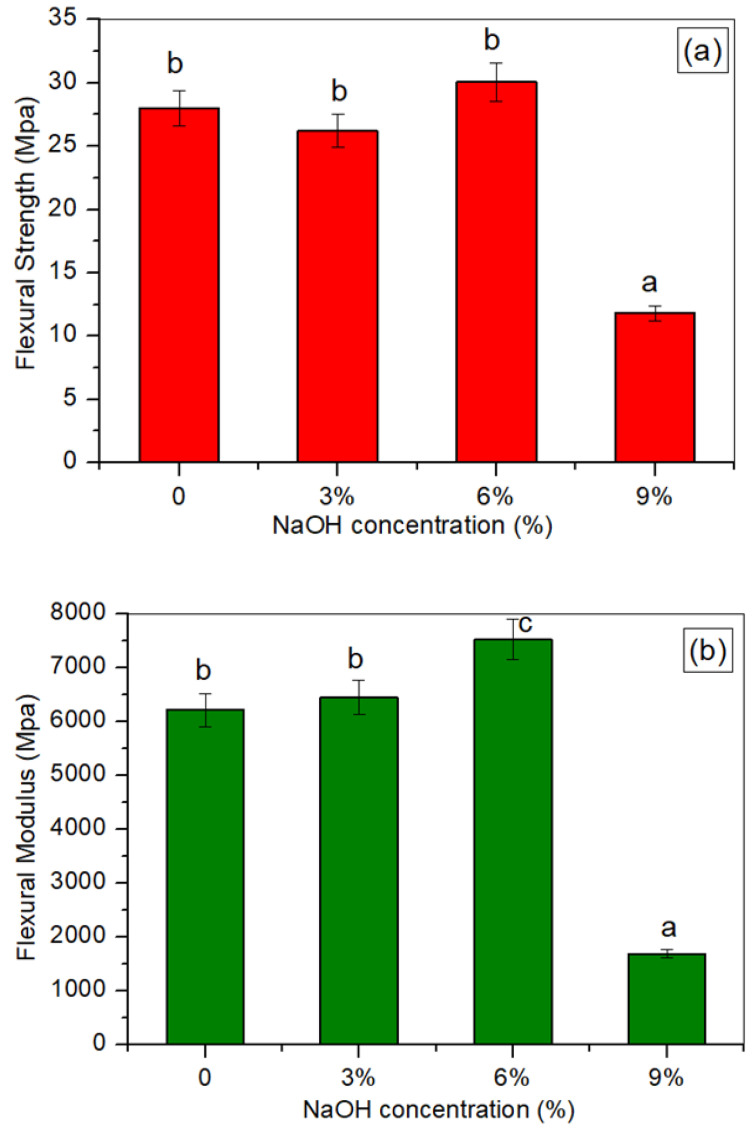
The (**a**) flexural strength, and (**b**) flexural modulus of untreated and treated TPCS/PW/CCF composites at different NaOH concentrations. Letters a, b, and c is indicates the group of data for statistical analysis.

**Figure 5 polymers-14-02769-f005:**
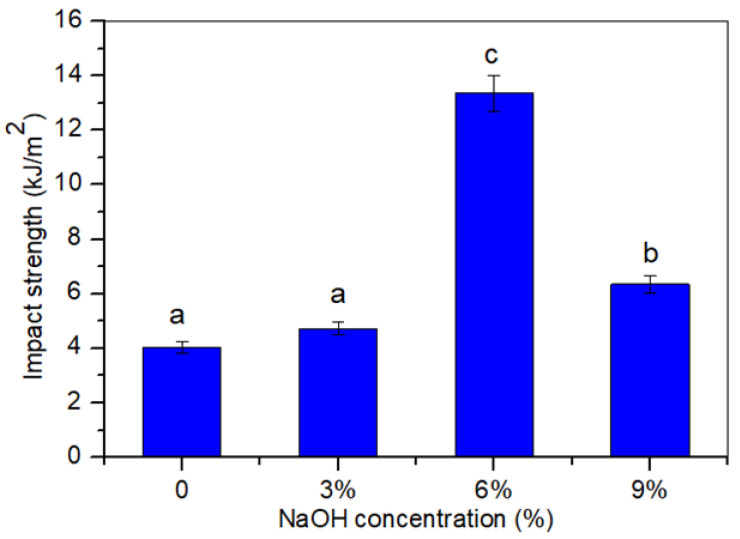
Impact strength of untreated and treated TPCS/PW/CCF composites at different NaOH concentrations. Letters a, b, and c is indicates the group of data for statistical analysis.

**Figure 6 polymers-14-02769-f006:**
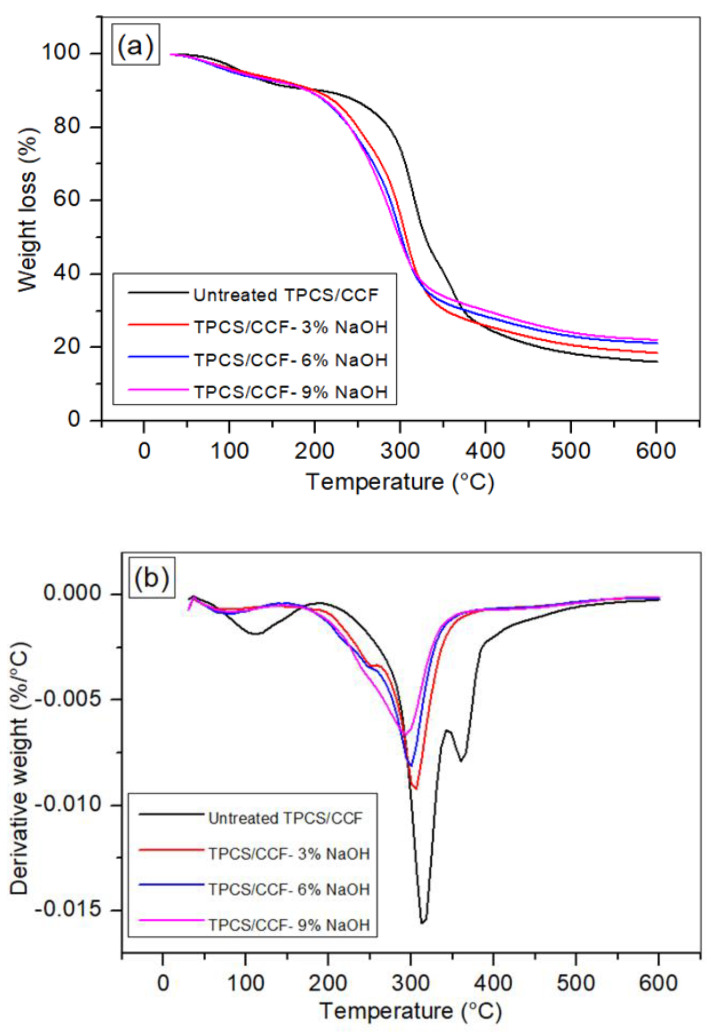
(**a**) TGA and (**b**) DTG of the untreated and NaOH-treated TPCS/PW/CCF composites.

**Figure 7 polymers-14-02769-f007:**
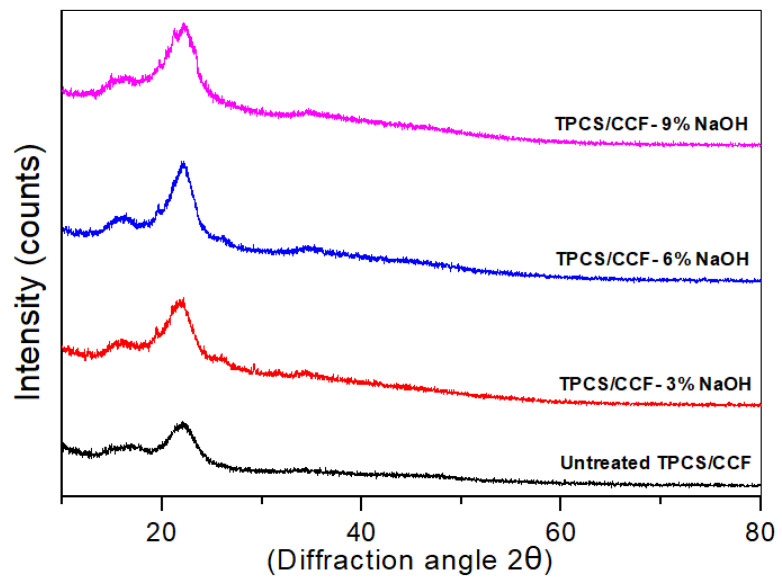
X-ray diffraction spectra of the untreated and treated TPCS/PW/CCF composites.

**Figure 8 polymers-14-02769-f008:**
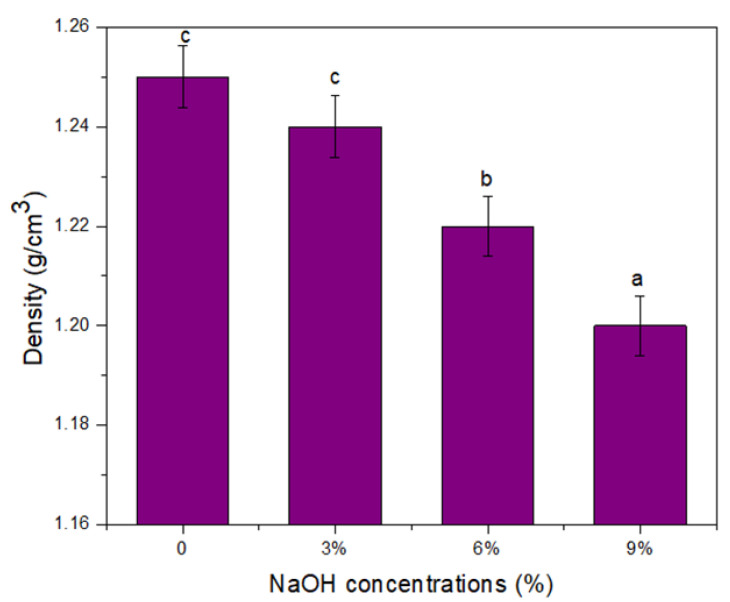
Density of untreated and treated TPCS/PW/CCF composites. Letters a, b, and c is indicates the group of data for statistical analysis.

**Figure 9 polymers-14-02769-f009:**
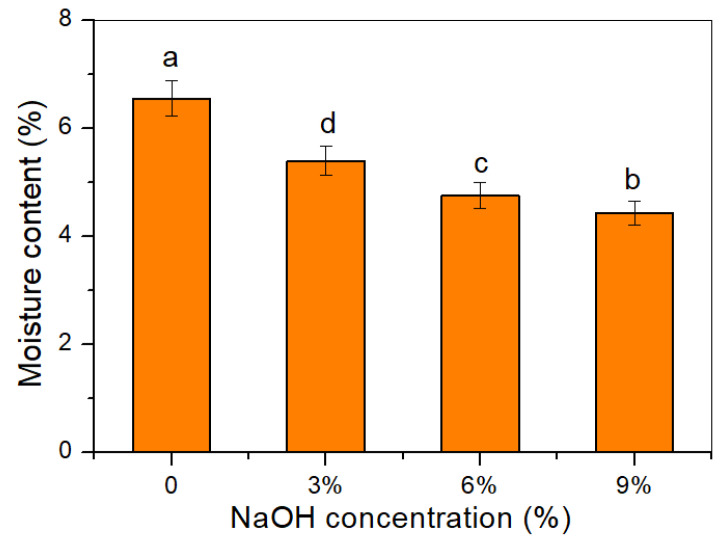
Moisture content of untreated and treated TPCS/PW/CCF composites. Letters a, b, c and d is indicates the group of data for statistical analysis.

**Figure 10 polymers-14-02769-f010:**
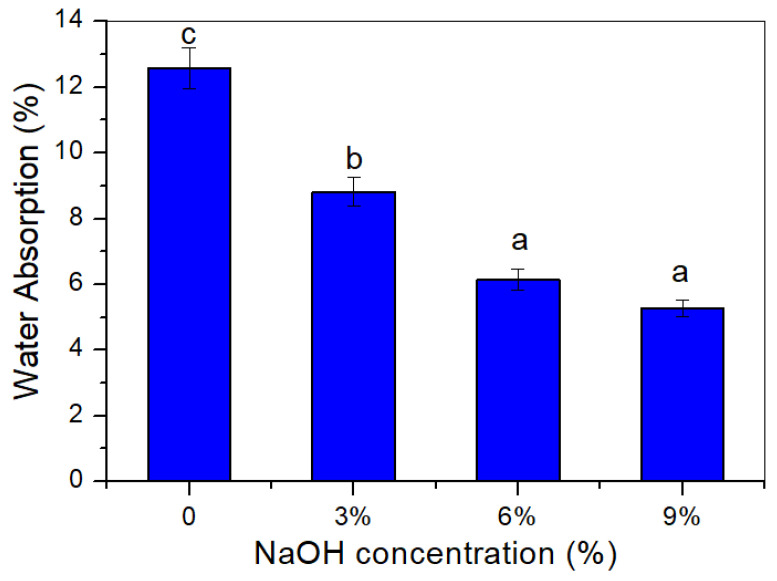
Water absorption of untreated and treated TPCS/PW/CCF composites. Letters a, b, and c is indicates the group of data for statistical analysis.

**Figure 11 polymers-14-02769-f011:**
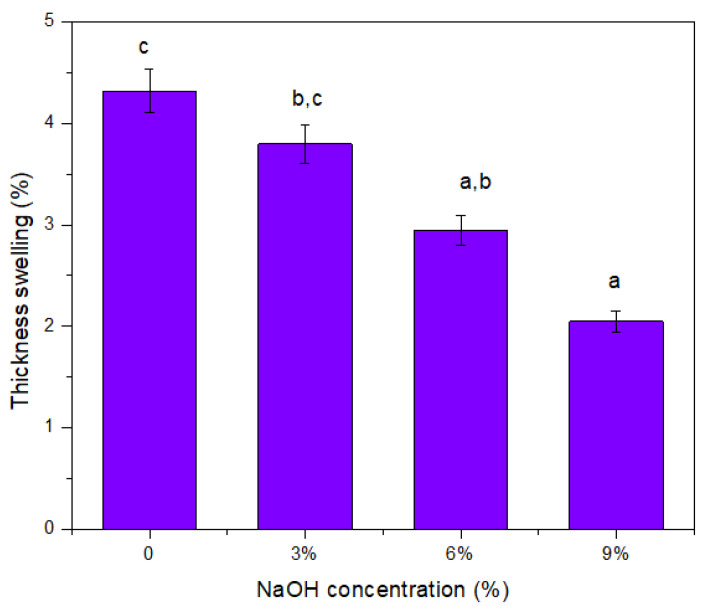
Thickness swelling of untreated and treated TPCS/PW/CCF composites. Letters a, b, and c is indicates the group of data for statistical analysis.

**Figure 12 polymers-14-02769-f012:**
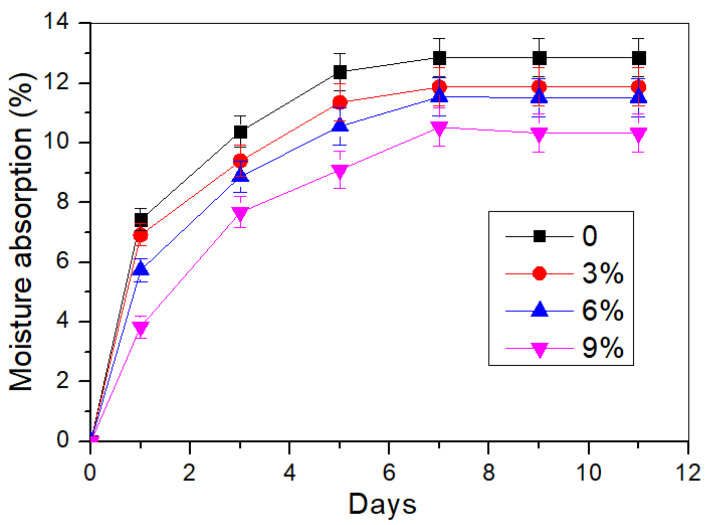
Moisture absorption of untreated and treated TPCS/PW/CCF composites.

**Figure 13 polymers-14-02769-f013:**
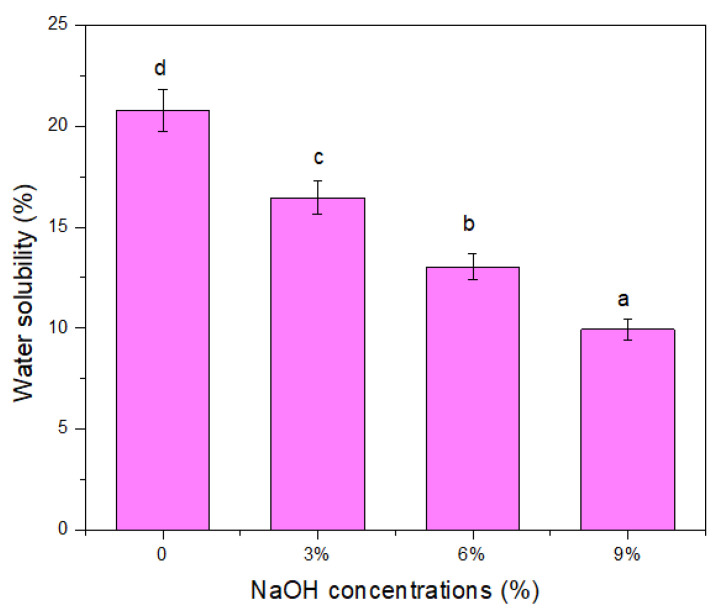
Water solubility of untreated and treated TPCS/PW/CCF composites. Letters a, b, c and d is indicates the group of data for statistical analysis.

**Figure 14 polymers-14-02769-f014:**
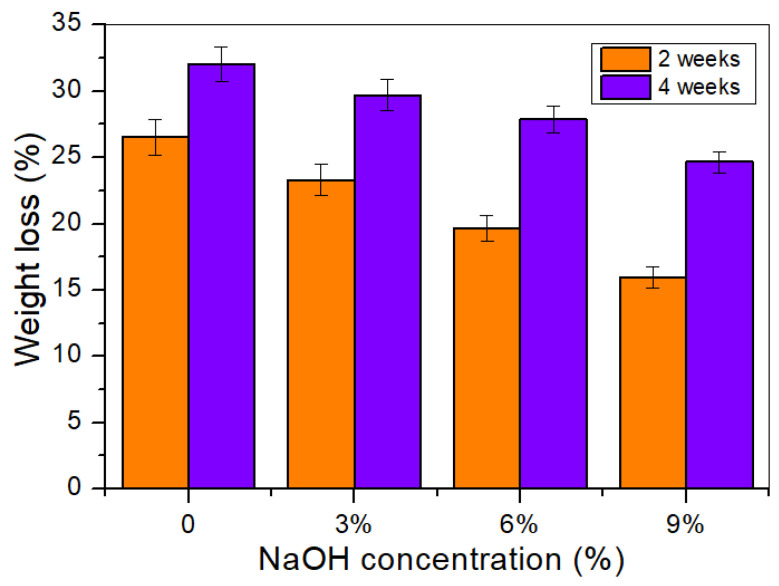
Weight loss of untreated and treated TPCS/PW/CCF composites after soil burial for two and four weeks.

**Table 1 polymers-14-02769-t001:** Analysis of variance (ANOVA) summary of TPCS/PW/CCF composites.

Variables	df	Flexural Strength	Flexural Modulus	Tensile Strength	Tensile Modulus	Elongation at Break	Impact
Mixture	3	0.00 *	0.00 *	0.00 *	0.00 *	0.00 *	0.00 *

Note: * Significantly different at *p* < 0.05.

**Table 2 polymers-14-02769-t002:** TGA and T_g_ results of treated and untreated TPCS/PW/CCF composites.

Samples	T_g_	T_on_	T_max_	Weight Loss at T_max_ (wt.%)	Char at 600 °C (wt.%)
	(°C)	(°C)	(°C)		
Untreated TPCS/PW/CCF	123.9	260	369	60.03	16.38
TPCS/PW/CCF - 3 wt.% NaOH	127.1	228	342	58.34	18.49
TPCS/PW/CCF - 6 wt.% NaOH	128.5	220	338	55.12	20.89
TPCS/PW/CCF - 9 wt.% NaOH	127.6	219	336	54.49	21.92

**Table 3 polymers-14-02769-t003:** Crystallinity index of untreated and treated TPCS/PW/CCF composites.

Samples	Crystallinity Index (%)
Untreated TPCS/PW/CCF	37.7
TPCS/PW/CCF - 3 wt.% NaOH	40.3
TPCS/PW/CCF - 6 wt.% NaOH	47.9
TPCS/PW/CCF - 9 wt.% NaOH	45.2

## Data Availability

The data presented in this study are available on request from the corresponding author.

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
