# Peer review of "Influence of Alkali Treatment on the Mechanical, Thermal, Water Absorption, and Biodegradation Properties of Cymbopogan citratus Fiber-Reinforced, Thermoplastic Cassava Starch–Palm Wax Composites"

_polymers, 2022, doi:10.3390/polym14142769_

Round 1
Reviewer 1 Report
The paper presents an interesting approach based on the Influence of Alkali Treatment on Mechanical, Thermal, Water Absorption and Biodegradation Properties of Cymbopogan Citratus Fiber Reinforced Thermoplastic Cassava Starch/Palm Wax Composites. However, the innovation of the current research work should be further highlighted and emphasized. At the same time, the authors should consider the following comments to greatly improve the quality of the paper.
1. There is a mix between UK-English and US-English since there are three times where fiber appeared as "fibre" in the manuscript. Kindly, search for them and replace them with fiber.
2. In the abstract, the final statement which says "These findings validated that alkaline treatment of CCF is able to improve the functionality of the fiber-reinforced composites. " requires speciality. Instead of generalizing the observation to all fiber reinforced composites, the statement has to specify it to Cymbopogan Citratus Fiber Reinforced composites.
3. The introduction needs to be improved by relating to the mechanics of the studied materials and their mechanical characteristics. The references to be included are: 10.1007/s10853-022-06994-3, 10.1177/0021998318790093, 10.1016/j.polymertesting.2017.09.009, 10.1177/07316844211051733, 10.1016/j.compstruct.2021.114698, 10.1177/0731684417727143 and 10.1002/app.46770.
4. In SEM analysis, this statement doesn't add any new information "The scanning electron microscope (SEM) is useful for performing microscopic investigation and characterization of fibers in terms of structural changes as well as surface morphology." So, it needs to be deleted. Also, what was the working distance?
5. The ASTM standards followed for the mechanical tests need to be referenced.
6. The tensile test setup and sample geometry need to illustrated with dimensions.
7. Figure 1 needs to have the same font style for x-axis and y-axis.
8. For SEM observations in Figure 2, is there any reason why the working distance for figure 2a was smaller than the the remaining figures?
9. the font style and size in x-axis and y-axis of Figure 5 need to be similar to the previous figures.
10. Table 1 has just null figures. The ANOVA analysis is incomplete in this table. Also, what does the symbol * refer to in that table?
11. The conclusion needs to be modified to summarize the research outcomes in short statements with clear observations.
Author Response
Dear Chief-Editor
Thanks for your letter and the thoughtful comments from the referees about our paper entitled “Influence of alkali treatment on mechanical, thermal, water absorption, and biodegradation properties of Cymbopogan citratus fiber reinforced thermoplastic cassava starch/palm wax composites” (ID: 1762222). We carefully analyzed all the comments and these comments are very valuable and helpful for perfecting and modifying our manuscript, and also have important guiding significance for our research. Therefore, we carefully checked the manuscript and revised it according to each comment. Consequently, we feel that our manuscript is substantially strengthened. Revised portion are marked using green background in the revised manuscript. The detailed corrections in the paper and the responses to the reviewer’s comments are as following list of revisions. We look forward to your positive response. If you have any question about this paper, please don’t hesitate to let us know. We hope these revisions will make it more acceptable for publication. Thank you.
Sincerely,
R. Jumaidin

Reviewer 2 Report
I have the following comments:
1. The aim is not clearly defined in the abstract.
2. Line 65: the reference is missing.
3. Line 74: the reference is missing.
4. The manuscript should contain some information about alternative plastic materials. The following reference should be used: Dordevic, D., Necasova, L., Antonic, B., Jancikova, S. and Tremlová, B., 2021. Plastic cutlery alternative: Case study with biodegradable spoons. Foods, 10(7), p.1612.
5. How FTIR analysis was conducted, it should be explained more in detail.
6. Table 2: there is no statistical analysis.
7. Table 4: there is no statistical analysis.
Author Response

(The authors gave the same response as above.)
